# Biosynthesis of the antibiotic nonribosomal peptide penicillin in baker's yeast

Ali R. Awan[1,2], Benjamin A. Blount[1,2], David J. Bell[1,3], William M. Shaw[1,2], Jack C.H. Ho[1,4], Robert M. McKiernan[1,4] & Tom Ellis[1,2]

Fungi are a valuable source of enzymatic diversity and therapeutic natural products including antibiotics. Here we engineer the baker's yeast *Saccharomyces cerevisiae* to produce and secrete the antibiotic penicillin, a beta-lactam nonribosomal peptide, by taking genes from a filamentous fungus and directing their efficient expression and subcellular localization. Using synthetic biology tools combined with long-read DNA sequencing, we optimize productivity by 50-fold to produce bioactive yields that allow spent *S. cerevisiae* growth media to have antibacterial action against *Streptococcus* bacteria. This work demonstrates that *S. cerevisiae* can be engineered to perform the complex biosynthesis of multicellular fungi, opening up the possibility of using yeast to accelerate rational engineering of nonribosomal peptide antibiotics.

[1] Centre for Synthetic Biology and Innovation, Imperial College London, London SW7 2AZ, UK. [2] Department of Bioengineering, Imperial College London, London SW7 2AZ, UK. [3] SynbiCITE Innovation and Knowledge Centre, Imperial College London, London SW7 2AZ, UK. [4] Department of Life Sciences, Imperial College London, London SW7 2AZ, UK. Correspondence and requests for materials should be addressed to T.E. (email: t.ellis@imperial.ac.uk).

Many important therapeutics including key antibiotics are derived from compounds produced by fungal organisms[1], yet fungal enzymatic diversity is a largely untapped resource for cheap biosynthesis of medical molecules and the potential for the discovery of novel therapeutics[2,3]. The baker's yeast *Saccharomyces cerevisiae* is the most well-characterized unicellular fungus and is also one of the main organisms used industrially for engineered biosynthesis[4]. Since the advent of synthetic biology, *S. cerevisiae* has been genetically reprogrammed to express diverse enzymes from bacteria and eukaryotes, producing hundreds of different molecules of industrial and therapeutic relevance including opiates and anti-malarial terpenoids[5–9]. A major class of bioactive molecules found in fungi as well as in bacteria are the non-ribosomal peptides (Nrp), a diverse set of complex molecules produced by large assembly-line enzymes called non-ribosomal peptide synthetases (NRPS). Non-ribosomal peptides include many front-line antibiotics including the classic beta-lactam antibiotic penicillin, which is naturally made by filamentous fungi[10,11].

To provide the foundations for using synthetic biology to produce fungal-derived Nrp molecules in a tractable host, we set out here to demonstrate that cells can be engineered to make the benchmark therapeutic penicillin. Despite a wide range of known fungal NRPS genes and the plethora of advanced genetic tools for *S. cerevisiae*, there have been no reports of engineered beta-lactam antibiotic production from this yeast. As is the case for most bioactive Nrp molecules, the biosynthesis of penicillin requires not only expression of the NRPS but also the coordinated expression of a pathway of tailoring enzymes that convert the Nrp into its active form. Previously, Siewers *et al.*[12] showed that co-expression of the *Penicillium chrysogenum* NRPS gene *pcbAB* and an NPRS activator gene in *S. cerevisiae* led to cytosolic synthesis of amino-adipyl-cysteinyl-valine (ACV), the Nrp intermediate in the five gene *P. chrysogenum* pathway of penicillin G (benzylpenicillin) biosynthesis. While this revealed that the first step of beta-lactam production was possible in *S. cerevisiae*, the pathway to producing an active Nrp antibiotic was incomplete. Using modular genetic tools to control enzyme expression, we demonstrate here that correct subcellular localization, along with expression optimization of the full five gene fungal pathway enables *S. cerevisiae* to synthesize benzylpenicillin. We further demonstrate that bioactive benzylpenicillin is secreted by our engineered yeast and that spent culture media has antibiotic activity against *Streptococcus* bacteria enabling a simple biological assay for productive strains.

## Results

**Establishing benzylpenicillin biosynthesis in *S. cerevisiae*.** The benzylpenicillin pathway in *P. chrysogenum* consists of five genes converting cysteine, valine and the non-canonical amino acid alpha-aminoadipic acid into a beta-lactam antibiotic via ACV (Fig. 1a). The 11.3 kbp NRPS gene *pcbAB* and the NRPS activator gene *npgA* required to produce ACV were first integrated into the *S. cerevisiae* BY4741 *TRP1* locus by inserting the two genes and the bidirectional GAL1/GAL10 promoters from the previously described pESC-npgA-pcbAB plasmid[12]. ACV production in this strain (Sc.A1) was then confirmed using liquid chromatography mass spectrometry (LCMS) comparing to a chemical standard (Fig. 1b). To complete the pathway and establish benzylpenicillin biosynthesis in *S. cerevisiae*, we next required efficient expression of the remaining three *P. chrysogenum* enzymes. However, as the final two steps of this pathway are known to naturally occur in the peroxisome in *P. chrysogenum*[13], we reasoned that cytoplasmic expression of all enzymes would not suffice and

that subcellular localization would be the key to full synthesis. We therefore took the step of tagging both *pclA* and *penDE* with the previously characterized *S. cerevisiae* peroxisome targeting sequence (PTS1) tag[14] and removing the native *P. chrysogenum* PTS1 tags to direct these enzymes to co-localize in the peroxisome on translation. To verify expression and correct subcellular localization of these two enzymes, we used fluorescence tagging and microscopy to show that a known *S. cerevisiae* peroxisomal protein (CIT2) co-localizes with both *pclA* and *penDE* tagged with *S. cerevisiae* PTS1 (Supplementary Fig. 1). Furthermore, variants of pclA and penDE harbouring the native *P. chrysogenum* peroxisome targeting tag did not localize to *S. cerevisiae* peroxisomes and peroxisomal localization for *CIT2*, *pclA* and *penDE* was abolished in a strain in which the PTS1-recognizing peroxisome importer *pex5* was deleted (Supplementary Fig. 1).

To test whether simultaneous expression of all five genes could result in benzylpenicillin production, we transformed a plasmid harbouring pcbC and the *S. cerevisiae* PTS1-tagged *pclA* and *penDE* into the Sc.A1 strain. The resulting strain (Sc.P1) was shown to indeed produce very low quantities of benzylpenicillin when measured by LCMS (Fig. 1c). By contrast, a strain harbouring *pclA* and *penDE* with *P. chrysogenum* peroxisome targeting sequences (Sc.P1x) produced ACV but failed to produce any benzylpenicillin (Fig. 1c). Finally, while the ACV Nrp intermediate was detected at equal amounts inside and outside cells, benzylpenicillin was found predominantly in supernatant fractions (at $\sim 90$ pg ml$^{-1}$), suggesting active secretion of the antibiotic (Fig. 1d).

**Combinatorial pathway optimization and nanopore sequencing.** Considering the small amounts of benzylpenicillin produced by the first engineered strain Sc.P1, we next sought to optimize production of benzylpenicillin in *S. cerevisiae* to achieve bioactive concentrations. One common strategy for improving biosynthesis yields is to alter the expression levels of the pathway enzymes, typically by changing the promoters for the corresponding genes[15]. This can increase the flux through the pathway preventing the build-up of any inhibitory intermediates[16] and can also aid in finding efficient expression levels of enzymes that allow their correct folding and subcellular localization, both important considerations for this case. Using a recently described modular cloning toolkit for yeast[17] and Golden Gate combinatorial DNA assembly[18] we constructed and tested hundreds of different combinations of the benzylpenicillin pathway genes with different promoters known to vary in strength and expression dynamics (Fig. 2, Supplementary Fig. 2). We used this approach to first optimize the production of the ACV Nrp intermediate (Supplementary Fig. 2), and then the conversion of ACV to benzylpenicillin (Fig. 2).

For the optimization of ACV production, we cloned the two genes for ACV biosynthesis with different pairs of promoters on a low-copy centromeric plasmid and measured ACV yields from transformed BY4741 yeast by LCMS. All strains constructed outperformed our Sc.A1 strain ($\sim 20$ ng ml$^{-1}$) and one combination, *pcbAB* with pTDH3 promoter and *npgA* with pPGK1 promoter, generated a strain (Sc.A2) that even outperformed yeast with the high-copy number pESC-npgA-pcbAB 2-micron plasmid containing strong galactose-inducible promoters ($\sim 280$ versus $\sim 70$ ng ml$^{-1}$; Supplementary Fig. 2, Supplementary Table 1).

To optimize conversion of ACV to benzylpenicillin, we next exploited one-pot combinatorial DNA assembly using Golden Gate cloning to make a diverse library of high copy plasmids in which genes *pcbC*, *pclA* and *penDE* are each expressed from one

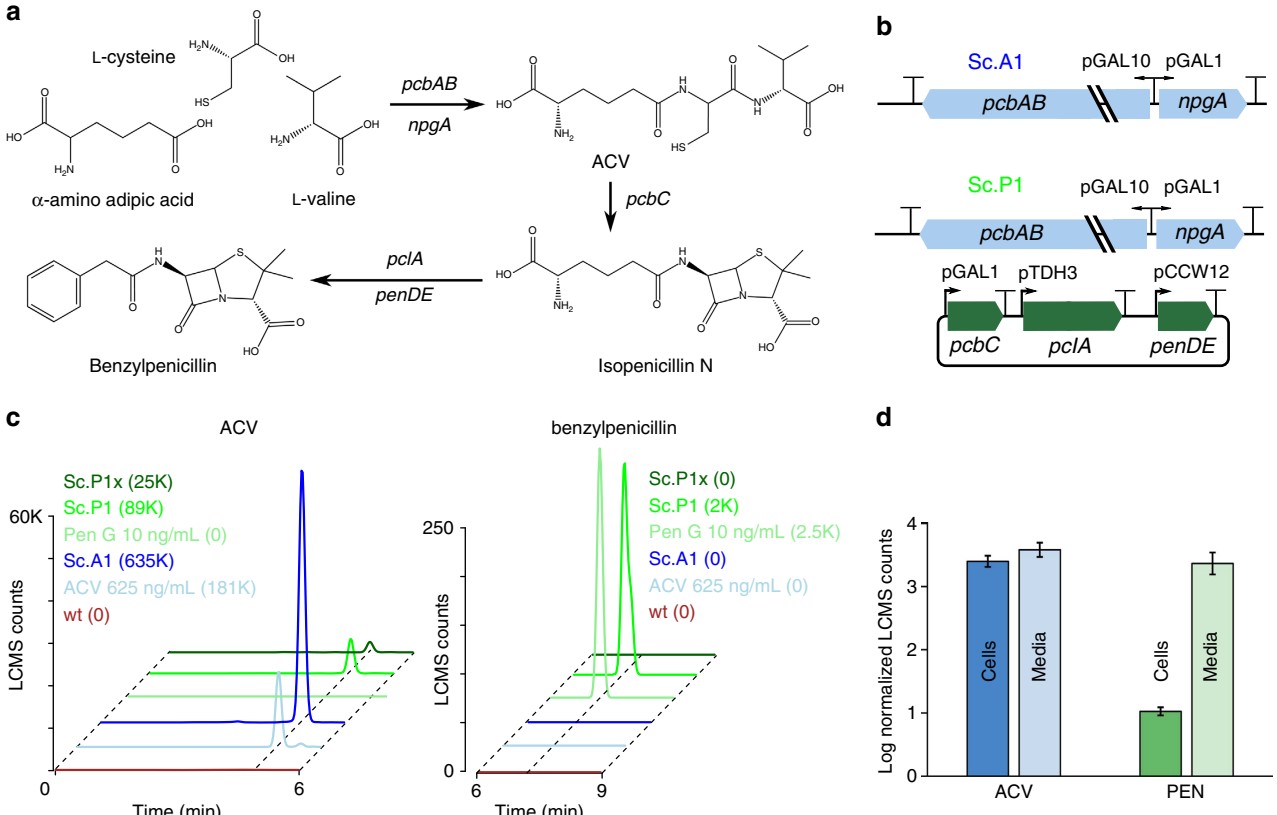

**Figure 1 | Production of benzylpenicillin by engineered *S. cerevisiae*.** (**a**) Benzylpenicillin biosynthesis requires a five-enzyme pathway where alpha-aminoadipic acid, cysteine and valine are first converted to the tripeptide ACV by *pcbAB* and *npgA*. ACV is then converted to benzylpenicillin by *pcbC*, *pclA* and *penDE*. (**b**) Strain Sc.A1 (*S. cerevisiae* BY4741 with genomically integrated *pcbAB* and *npgA* genes) produces ACV as observed by LCMS analysis. Parallel diagonal lines in the cartoon for the *pcbAB* gene represent that this very large gene is not drawn to scale with *npgA*. To produce benzylpenicillin, strain Sc.A1 was transformed with a plasmid expressing *pcbC*, *pclA* and *penDE* to make strain Sc.P1. (**c**) LCMS counts versus retention time is shown for the detection of ACV (left) and benzylpenicillin (right), for an ACV standard (light blue) and a benzylpenicillin standard (light green) as well as for supernatant from a wild-type culture (brown), an Sc.A1 culture (blue), an Sc.P1 culture (green) and a culture of strain without peroxisomal targeting of *pclA* and *penDE* (Sc.P1x, dark green). All cultures were grown at 20 °C in the presence of 5 mM AAA and 0.25 mM phenylacetic acid (required for the last step of benzylpenicillin biosynthesis). Dashed diagonal lines indicate the bounds of the x axis and the centres of the relevant peaks. Rounded LCMS counts of the tallest peak for both ACV and benzylpenicillin detection are indicated on the y axis, and areas under each peak are given in parentheses in the legend. Mass spectra are included as Supplementary Data 2. (**d**) Benzylpenicillin is secreted as shown by concentration-normalized $\log_{10}$ values of LCMS counts for cell pellets and media for both ACV and benzylpenicillin from Sc.P1 cultures grown as in **c**. Error bars show s.d. for three biological replicates.

of ten randomly assigned promoters that span a range of strengths (Fig. 2a). Eight different constitutive promoters and two galactose-inducible promoters were included. The constitutive promoters were classified as 'strong' (4 total), 'medium' (2 total) or 'weak' (2 total) promoters according to their characterization in a previous study (Supplementary Table 7)[17]. The resultant plasmid library, with a theoretical diversity of 1,000 members, was transformed into the Sc.A2 strain and a total of 160 colonies were individually screened by LCMS for benzylpenicillin production in Synthetic Complete medium minus the appropriate amino acids for selection, and with glucose as the carbon source (Supplementary Table 2).

From the screened colonies, the plasmids from the top 10 out of 20 strains showing detectable production of benzylpenicillin by LCMS (Supplementary Table 2) were sequenced to determine the promoter combinations for *pcbC*, *pclA* and *penDE* that direct efficient biosynthesis in *S. cerevisiae*. Sequencing revealed an apparent over-representation of strong constitutive promoters at the *pclA* gene (10 out of 10) and of medium constitutive promoters at the *pcbC* gene (5 out of 10) in strains with benzylpenicillin production (Fig. 2b, Supplementary Table 3).

These results suggested that either a strong level of *pclA* expression is required for optimal benzylpenicillin production in *S. cerevisiae*, or that our combinatorial DNA assembly had an unintended bias for incorporation of these promoter parts into the final plasmid library. To rule out any bias in our library assembly, we used a MinION DNA sequencer (Oxford Nanopore Technologies) to provide long-read sequencing of the assembly products pooled from all stages in our plasmid library construction (Fig. 2a). The ability of nanopore sequencing to routinely return read lengths above 5 kb make it well suited to determining the full-length products of modular DNA construction[19]. Two-directional (2D) reads of products from multigene assembly revealed *in vitro* construction of over one hundred plasmids containing different combinations of promoters in front the three pathway genes. No significant bias for incorporation was seen during either of the DNA assembly steps compared to the distribution expected by chance (Fig. 2c, Supplementary Table 3, $\chi^2$ test), whereas a clear bias was seen for strong promoters associated with *pclA* in all yeast strains able to perform detectable benzylpenicillin biosynthesis (Fig. 2c, Supplementary Table 3, Fisher's exact test).

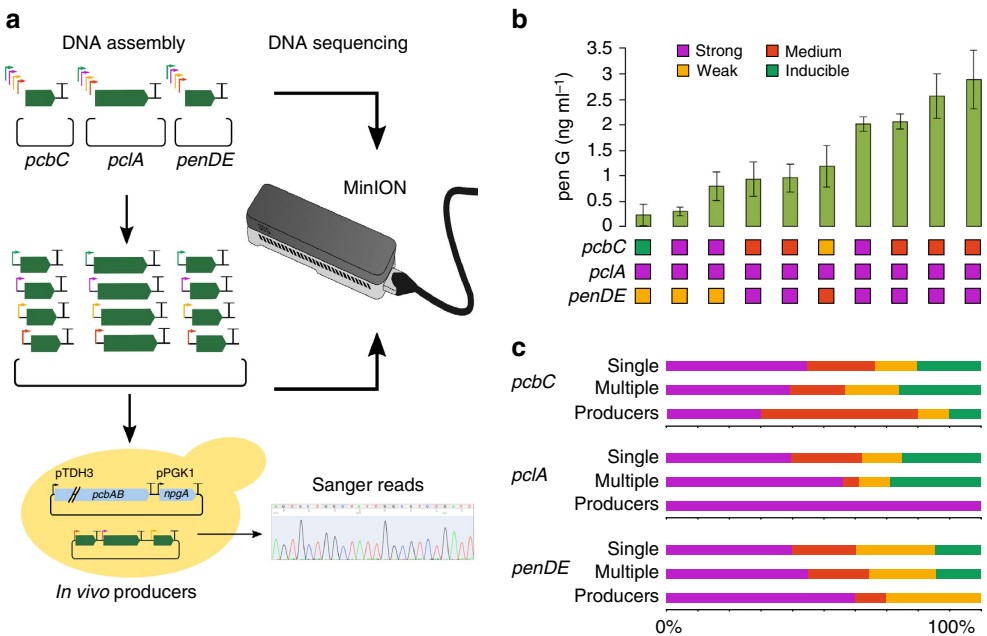

**Figure 2 | Optimization of benzylpenicillin production in *S. cerevisiae*.** (**a**) Construction and sequencing of a library with alternative promoters for *pcbC*, *pclA* and *penDE*. Ten different promoters of four categories (strong, medium, weak or inducible) were randomly used to drive expression of each of the genes responsible for conversion of ACV to benzylpenicillin. Plasmid libraries of single genes with different promoters were initially constructed and then pooled to randomly assemble a multigene plasmid library. Single gene and multigene libraries were subjected to MinION nanopore sequencing. The multigene library was transformed into ACV producer Sc.A2 and transformants producing benzylpenicillin were Sanger sequenced to reveal identity of promoters. (**b**) Concentrations of benzylpenicillin secreted into the supernatant by the ten yeast strains selected from the promoter screen. The best producing strain makes ~3 ng ml$^{-1}$ of benzylpenicillin as determined by LCMS. Under each bar, the category of the promoter in front of each gene is shown. Error bars represent s.e.m. from three biological replicates. (**c**) Comparison of the percentage of promoters from each of the four categories for each pathway gene during Golden Gate library construction (single and multiple gene steps) and among the ten benzylpenicillin producing yeast strains as determined by DNA sequencing.

**Antibiotic action of the spent yeast culture media.** Optimization of the biosynthesis pathway through combinatorial cloning with alternative promoters led to a ~30-fold increase in yields of secreted benzylpenicillin compared to our first strain Sc.P1 (~90 pg ml$^{-1}$). The highest producer, strain Sc.P2, secretes benzylpenicillin into its growth media at a concentration calculated to be around 3 ng ml$^{-1}$ (Fig. 2b). Using this strain, we next sought to test whether benzylpenicillin secreted by engineered *S. cerevisiae* shows expected bioactivity. The secreted levels of benzylpenicillin are similar to those required for antibiotic action against various bacteria such as *Streptococcus pyogenes*[20]. Therefore, to test the bioactivity of the secreted benzylpenicillin, we diluted liquid cultures of *S. pyogenes* with spent culture medium from either Sc.P2 or from an inactivated version of Sc.P2 (Sc.P2x) that has a premature termination codon in *pcbAB* and does not produce benzylpenicillin (Fig. 3a). Overnight growth of *S. pyogenes* was almost completely inhibited in the presence of spent culture media from Sc.P2. By contrast, spent culture media from the inactivated Sc.P2x strain resulted in normal growth of *S. pyogenes* (Fig. 3b). Simultaneous exposure of *S. pyogenes* to a range of concentrations of benzylpenicillin standards revealed that the spent culture media from Sc.P2 had the antibiotic action equivalent to 3 ng ml$^{-1}$ of benzylpenicillin being present. This concentration was then verified in parallel by LCMS analysis of the Sc.P2 culture media, demonstrating that the benzylpenicillin secreted by *S. cerevisiae* has the same bioactivity as the commercially obtained standard.

To demonstrate the utility of this simple bioactivity assay, we next employed it to further explore pathway expression optimization by exchange of *pcbC*, *pclA* and *penDE* promoters.

The finding from the previous screen (Fig. 2) that stronger promoters seemed to be favoured at the *pclA* and *penDE* genes for high benzylpenicillin yields led us to test a new promoter library especially enriched for stronger promoters (Supplementary Table 10). Specifically, we constructed a combinatorial library of strains harbouring the pcbAB-npgA plasmid from Sc.A1 as well as different variants of a second plasmid bearing the *pcbC*, *pclA* and *penDE* genes, each driven by one of six different promoters ranging from medium to strong. We screened 120 different strains over two 96-well plates and selected 12 strains that exhibited a range of *S. pyogenes* inhibition (Fig. 3c). To verify the bioactivity screen, we validated benzylpenicillin yields of these strains by LCMS (Fig. 3d) revealing a clear relationship between the concentration of the secreted antibiotic and the inhibition of bacterial growth. From this new screen, five strains gave higher benzylpenicillin yields than Sc.P2 when measured by LCMS, with two giving yields of >5 ng ml$^{-1}$, a more than a 50-fold increase in yield relative to our original producer strain, Sc.P1 (~90 pg ml$^{-1}$). Sequence analysis of the three strains secreting the highest yields (Fig. 3e) revealed no clear pattern of use among the strong promoters whose expression strengths cover an order of magnitude (Supplementary Table 12). The exact relationship between benzylpenicillin production and the promoter strengths of the *pcbC*, *pclA* and *penDE* genes is likely to be more complex and could be further explored in the future by metabolic flux analysis approaches[21]. Encouragingly, the results from our screen illustrate how a simple assay based on antibiotic activity can be used to quickly and cheaply screen hundreds of strains for productivity without the need for LCMS.

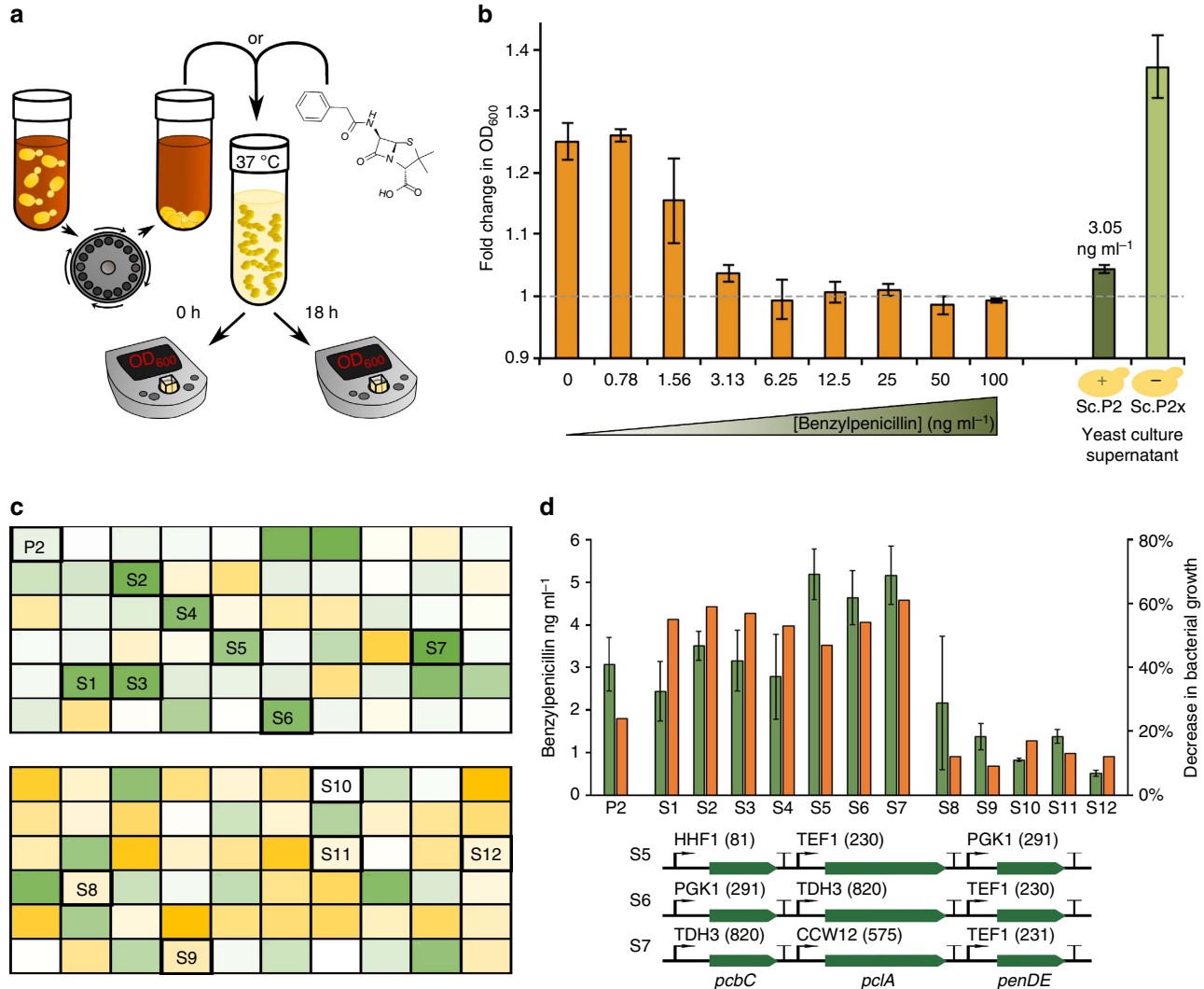

**Figure 3 | Secreted benzylpenicillin is a bioactive antibiotic. (a)** Schematic of the spent culture media antibiotic activity test. (**b**) Fold change in OD600 of *S. pyogenes* cell cultures grown either in the presence of increasing concentrations of benzylpenicillin standard over 18 h (orange bars, left) or in the presence of supernatant from benzylpenicillin producing yeast Sc.P2 (dark green bar, plus sign) and yeast Sc.P2x with an inactive *pcbAB* gene (light green bar, minus sign). Concentration of benzylpenicillin in Sc.P2 supernatant as measured by LCMS is given above the dark green bar. Error bars show s.e.m. for three biological replicates. (**c**) Two 96-well plates of *S. pyogenes* were grown in the presence of supernatant from *S. cerevisiae* strains from the second promoter screen and the top strain from the first promoter screen (Sc.P2, abbreviated to P2). OD600 readings were taken at time zero and again after 18 h of incubation at 37 °C. Heatmaps indicate the degree of *S. pyogenes* overnight growth with green showing low OD600 (inhibtion of growth) and orange showing high OD600. (**d**) The percentage of inhibition of *S. pyogenes* growth from the screen (orange) and the LCMS-determined benzylpenicillin yields (green) are shown for spent growth media from Sc.P2 and 12 selected strains (seven showing high growth inhibition, five showing low growth inhibition). Error bars represent s.e.m. from three biological replicates. The promoters and their relative strengths (a.u.) used to express the three pathway genes in the three strains showing the highest benzylpenicillin production are shown underneath the chart.

## Discussion

Our results demonstrate, to the best of our knowledge, the first biosynthesis of a bioactive Nrp antibiotic from engineered *S. cerevisiae* yeast. After achieving production of benzylpenicillin, we improved yields by optimizing pathway enzyme expression and exploited long-read nanopore sequencing to verify the combinatorial DNA assembly of our libraries to ascertain the best combination of promoter strengths for increased yields. We observed that this required strong expression of the peroxisomally located enzyme *pclA*. As engineered heterologous biosynthesis of penicillin has only previously been observed with *Hansenula polymorpha*, a methylotrophic yeast with large peroxisomes[22], this suggests that boosting peroxisomal expression further, for example by increasing peroxisome formation, by may be a future approach to improve yields from our *S. cerevisiae*, which

are currently three orders of magnitude below *P. chrysogenum* yields and two orders of magnitude below yields achieved in *H. polymorpha*[22]. Alternatively, further yield improvements could also be achieved by using other metabolic engineering strategies[23,24]. However, as existing production strategies already achieve sufficient yields of penicillin for global medical needs, increasing the productivity of our engineered *S. cerevisiae* strains is not a priority.

Instead, we view our achievement in *S. cerevisiae* as a paradigm for establishing complex fungal biosynthesis pathways in a standard chassis organism used extensively in research and industry. Given the impressive wealth of genetic tools and the ease of engineering with this microbe, we see it as the ideal testbed for screening and diversifying fungal pathway biosynthesis. Thus, the real utility of our work is to lay the foundations for

future engineering projects that can mix and match heterologous NRPS enzymes with tailoring enzymes. We especially anticipate that expression and genetic manipulation of other bioactive Nrp pathways in *S. cerevisiae* could advance the screening of engineered or newly discovered NRPS enzymes from fungi. NRPS enzymes represent a particularly attractive class of enzymes for engineering efforts as they are inherently modular with each module region recognizing one amino acid and incorporating it into the Nrp product[15,24]. NRPS modules incorporate both the standard 20 amino acids, as well as hundreds of non-proteinogenic amino acids, including D-enantiomers[25]. Thus by combining different modules together it should be possible make chimeric NRPS enzymes that produce thousands of novel Nrp molecules predisposed by evolution for bioactivity[26]. However, determining the structural boundaries of modules within natural NRPS enzymes has so far proved extremely challenging due to the difficulty in obtaining structures for these massive enzymes[27,28]. An alternative approach to structure-guided mutagenesis is exhaustive mutation and screening. Simple biological screens such as those based on the antibiotic activity shown in this study could help pave the way towards true combinatorial biosynthesis of novel bioactive Nrp molecules. Such endeavors are extremely important in light of widespread resistance to antibiotics and the decreasing number of new antibiotics developed through traditional means[29].

## Methods

**Construction of strains.** Strain Sc.A1 was made by replacing the TRP1 gene in BY4741 with the *pcbAB-npgA* segment from the pESC-npgA-pcbAB plasmid from a previous study[12]. A URA3 gene from *K. lactis* was integrated upstream of the *pcbAB* gene to allow genomic integration and the use of media without uracil was used to enable comparison of the ACV production of this strain with that of a BY4741 strain transformed with the ura-marked pESC-npgA-pcbAB plasmid. A full genetic map of the altered TRP1 locus is provided as an annotated Genbank file in Supplementary Data 1. The PEX5 deletion strain was constructed by CRISPR-enhanced recombination. Linear fragments encoding Cas9, a gRNA retargeted to PEX5 and an overlap extension PCR product encoding a whole-CDS deletion of PEX5 were co-transformed into BY4741 following the protocol outlined at benchling.com/pub/ellis-crispr-tools. Oligonucleotides used retarget the gRNA and generate the deletion template PCR product are listed in the Supplementary Information. To change the native *P. chrysogenum* Peroxisome Targeting Sequences (PTS1) to those of *S. cerevisiae* for the *pclA* and *penDE* genes, the native C-terminal tripeptide SKI for *pclA* and ARL for *penDE* were both changed to the *S. cerevisiae* PTS1 tripeptide SKL.

All other strains used in all experiments were constructed by transforming plasmids into BY4741 or into the *Δpex5* strain. These are specified in Supplementary Table 4. Annotated Genbank files of all plasmids are provided in Supplementary Data 1.

**Growth of strains for ACV and penicillin production.** For all ACV and penicillin producing experiments, cultures were prepared in the following manner. After initial construction, strains were stored in 25% glycerol stocks at −80 °C. For recovery of strains from glycerol stocks, strains were streaked onto the appropriate selective media agar plates and incubated at 30 °C for 2–3 days. Single colonies were picked using a pipette tip and used to innoculate 4 ml overnight cultures in synthetic complete media minus the appropriate amino acids for selective pressure with either glucose or galactose as the carbon source. There were no secondary precultures. For plate based assays, cells were grown overnight at 700 r.p.m. at 30 °C. For 50 ml falcon tube based assays, cells were grown at 225 r.p.m. at 30 °C. Overnight cultures were then back-diluted into production media and grown at 20 °C (216 r.p.m.) for 20 h (for plate based assays) or until the OD600 reached between 0.6 and 0.8 (for 50 ml falcon tube assays). Supplementary Table 5 details the composition of production media for different experiments.

**Fluorescence microscopy.** Microscopy for Supplementary Fig. 1 was carried out with a Nikon Eclipse Ti, using the NIS Elements AR software. The objective was set at ×60. Slides were fixed with yeast cells to visualize. The excitation wavelengths for detection of Venus, mRuby2 and mTurquoise2 fluorescence were 535, 590 and 535 nM, respectively.

**Preparation of standards and samples for LCMS.** For all LCMS experiments, standards were prepared as follows. ACV standards were prepared by dissolving ACV (BACHEM H-4204) in water to a concentration of 10 ng μl⁻¹ and

making three 10-fold dilutions. This gave four standards with concentrations of 10 ng μl⁻¹, 1 ng μl⁻¹; 100 pg μl⁻¹; 10 pg μl⁻¹. Benzylpenicillin standards were prepared by dissolving the sodium salt of penicillin G (Sigma P3032) in water. The same concentrations were used for benzylpenicillin as were used for ACV.

Cellular extracts for LCMS for the data in Fig. 1 were prepared as follows: 30 ml of cell culture was collected at an OD600 of 0.6. Cell culture was centrifuged at 7,000g for 10 min, and supernatant was either kept for LCMS (as in Fig. 1d) or discarded. The cell pellet was resuspended in 100 μl methanol. A volume of 50 μl of the resuspension was transferred to a microcentrifuge tube with 25 μl of glass beads (Sigma G8772-100G) on ice. The tube was then vortexed for 30 s and then placed on ice for 30 s, and these two steps were repeated three times (for a total of four sets of vortexing and incubation on ice). The tube was then centrifuged at 12,000g for 30 min, and 40 μl of supernatant was aliquoted to a separate tube for LCMS measurement. Supernatants for LCMS from cultures grown in 96 well plates in Fig. 2 were obtained by centrifuging plates at 3,000g for 30 min.

**Liquid chromatography mass spectrometry.** An LC/MS/MS method was developed for the measurement of ACV and benzylpenicillin, using an Agilent 1290 LC and 6550 quadrupole time-of-flight (Q-ToF) mass spectrometer with electrospray ionization (Santa Clara, CA). The LC column used was an Agilent Zorbax Extend C-18, 2.1 × 50 mm and 1.8 μm particle size. The LC buffers were 0.1% formic acid in water and 0.1% formic acid in acetonitrile (v/v).

The gradient elution method is detailed in Supplementary Table 6. Quantification was based on the LC retention times of standards and the area of accurately measured diagnostic fragment ion for each molecule (Supplementary Table 6). The protonated molecules of each analyte [M + H] +, were targeted and subjected to collision induced dissociation (collision energy 16 eV), with product ions accumulated throughout the analysis. Solutions of benzylpenicillin and ACV standards in water were used to generate calibration curves.

The linear range of the method was determined by injecting standards over a range of concentrations. The lower limit of detection was determined by the amount a sample resulting in a peak with a signal-to-noise of 3:1. The lower limit of quantification was taken to be the concentration of analyte that produced a signal-to-noise of 10:1. The lower limit of detection and lower limit of quantification for benzylpenicillin were found to be on-column injections of 5 and 20 pg, respectively. Further specifications are found in Supplementary Table 6. All mass spectra are included as Supplementary Data 2.

**Calculation of ACV and benzylpenicillin yield from LCMS data.** For both ACV and benzylpenicillin, pure chemical standards were run at the following concentrations: 10 ng ml⁻¹, 100 ng ml⁻¹, 10 μg ml⁻¹, 100 μg ml⁻¹. The corresponding LCMS counts for these standards were plotted against the concentrations of the standards and the linear range of the resulting plot was used to construct a line of best fit in excel. The corresponding line equation was used to obtain values for the yield in ng ml⁻¹ of ACV and benzylpenicillin from experimental samples based on the LCMS counts for these molecules.

**Promoter screens for optimizing ACV to penicillin conversion.** The assembly of multigene (*pcbC*, *pclA*, *penDE*) plasmids with ten randomized promoters (Supplementary Table 7) was split into two stages: assembly of single-gene constructs, then assembly of multigene constructs. For single-gene construct assembly, an equimolar mix of all ten promoters was made with a final concentration of 50 fM (referred to as 'promoter mix', Supplementary Table 9). This was used as a type 2 plasmid according to the yeast toolkit specification[17]. Then, Golden Gate reactions were set up with the following parts according to the Yeast Toolkit cassette plasmid golden gate assembly protocol (Supplementary Table 9).

Each of the three reactions were transformed into *E. coli*, and for each of the three transformation plates, transformant colonies were mixed together into a single overnight culture each. From each of the three overnight cultures a plasmid library was prepared, of which an aliquot was used to construct a pooled sample for nanopore sequencing (see nanopore sequencing section). The three resulting single-gene plasmid libraries were used to set up a single multigene golden gate reaction according to the yeast toolkit protocol (Supplementary Table 9).

This Golden Gate reaction was transformed into *E. coli*, and all transformant colonies were mixed together into a single overnight culture. A multigene plasmid library was prepared from this overnight culture. Part of this plasmid library was prepared for nanopore sequencing (see nanopore sequencing section), while 4 μg was used to transform into *S. cerevisiae* strain Sc.A2. The resulting transformants were screened by LCMS for the production of benzylpenicillin (Supplementary Table 2) and the promoter regions of the multigene plasmids from producer strains were identified by Sanger sequencing.

A second promoter screen was carried out to test the suitability of the *S. pyogenes* growth inhibition assay for identifying strains with improved benzylpenicillin yield. Strains were constructed in analogous fashion to that described above, but with different promoters (Supplementary Tables 10 and 11).

**Nanopore sequencing of library construction.** To enrich for the penicillin pathway assembly DNA and remove assembly vector backbone DNA, the

multigene assembly library was digested with EcoRI and AlwNI. Restriction digest products ranging from 5,616 to 6,117 bp were isolated by agarose gel electrophoresis and purified using a QIAquick gel extraction kit (Qiagen). The single pathway gene assembly libraries were similarly enriched by digestion with BsmBI and AlwNI. Restriction digest products ranging from 2,062 to 2,229, 2,549 to 2,716 and 1,885 to 2,052 bp for the *pcbC*, *pclA* and *penDE* assemblies, respectively, were purified.

Enriched assembly DNA for the multigene and single gene assemblies was quantified on a Qubit 2.0 fluorometer (Thermo Fisher Scientific) using a Qubit dsDNA HS Assay Kit (Thermo Fisher Scientific). The four samples were combined to a give an equimolar mix (assuming a molecular weight for each assembly based on the mean promoter length) with a total DNA content of 2.6 μg in 45 μl dH$_2$O.

DNA underwent end repair using NEBNext FFPE DNA Repair Mix (M6630, New England Biolabs) according to the manufacturer's instructions. The repaired DNA was recovered using Agincourt AMPure XP beads (A63880, Beckman Coulter), washed twice in 200 μl 70% ethanol and eluted in 46 μl dH$_2$O. DNA was then dA-tailed using NEBNext Ultra II End Repair/dA-Tailing Module (E7546, New England Biolabs) according to the manufacturer's instructions and recovered using Agincourt AMPure XP beads as before, eluting in 30 μl dH$_2$O.

The dA-tailed library DNA was then processed using Blunt/TA Ligase Master Mix (M0367, New England Biolabs) and a Nanopore Sequencing Kit (SQK-NSK007, Oxford Nanopore Technologies) according to the manufacturer's instructions to ligate adaptors and tethers to the library. 50 μl Dynabeads MyOne Streptavidin C1 beads were washed twice in buffer BBB (Nanopore Sequencing Kit) and then resuspended in 100 μl BBB. These beads were then added to the processed DNA sample and mixed for 5 min at room temperature. Beads were washed twice with 150 μl BBB before eluting the sample in 25 μl ELB (Nanopore Sequencing Kit). The library was quantified by Qubit as before and yielded a total of 253 ng.

**Nanopore sequencing.** A fresh MinION R7 Flow Cell Mk I (FLO-MIN104, Oxford Nanopore Technologies) was loaded into a MinION MK I (MIN-MAP002, Oxford Nanopore Technologies) and primed using a Nanopore Sequencing Kit according to the manufacturer's instructions. The sequencing mix was generated by combining 75 μl RNB, 65 μl NFW and 4 μl Fuel Mix (Nanopore Sequencing kit) before adding 6 μl of the processed DNA library. This sequencing mix was loaded into the flow cell and sequenced using the 48 h sequencing script on MinKNOW (Oxford Nanopore Technologies). After 18 h, the script was stopped and a fresh sequencing mix was prepared and loaded into the flow cell. The 48 h sequencing script was then restarted. This reloading process was repeated after a further 4.5 h. The sequencing script was stopped once read acquisition had slowed to less than 1 successful read in a 5 min period.

**Analysis of nanopore sequencing data.** Oxford Nanopore's cloud-based Metrichor application '2D Basecalling for SQK-MAP006 v1.69' was used to basecall data from a MinION run that used R7.3 chemistry. Poretools[30] (https://github.com/arq5x/poretools) was used to extract sequence files and the program lastal v658 (http://last.cbrc.jp/) was used to align the 2D reads to a database of all the potential promoters and all combined CDS + terminators using options -s2 -T0 -Q0 -a1 -fTAB -e50*. To remove reads corresponding to other DNA sequences present, the database was also populated with sequences for AmpRTerm_AmpR_AmpRProm (part of the single assembly plasmid backbone), His3Prom_His3Term_2Micron_KanRTerm_KanR_KanRProm (part of the multiple assembly plasmid backbone), ColE1 (part of both the single and multiple assembly plasmid backbone), and the lambda phage whole genome.

To build up a picture of which part of a plasmid each read represented, a custom-built script ordered alignments first by read, then by read coordinates. It then identified reads as originating from a particular plasmid by looking at plasmid regions in the database that each read had aligned to. These identified reads were required to be a similar length (within 15%) to the sum of lengths of regions of plasmids that they were aligned to. The script identified digested multiple gene assemblies, digested single gene assemblies and, due to reduced function of BsmBI, non-digested single assemblies. To be identified as a digested single gene assembly, the read was also required to start and finish within 50 bp of the start and end of the first and final regions respectively. This minimized the chance of misidentifying a multigene assembly as a single-gene assembly. The promoters at each position within these identified reads were recorded.

*[-s2 use both query strands. -T0 local alignment. -Q0 use fasta as input. -a1 gap existence cost of 1. -fTAB tabular output -e50 minimum gap alignment score of 50.]

***S. pyogenes* growth inhibition assays.** An overnight culture of *S. pyogenes* (H584, M1 type) was grown for 24 h at 37 °C (5% CO$_2$) in Todd Hewitt Broth. This culture was diluted in 10× Todd Hewitt Broth to an optical density of OD600 0.2. In separate wells of an optically transparent 96-well plate (VWR 3596) 10 μl of this solution was added to 90 μl of either a known concentration of benzylpenicillin dissolved in 0.25 mM phenylacetic acid or 90 μl of spent culture media from Sc.P2 or Sc.P2x or other potential benzylpenicillin-producing strains grown in production conditions. Sc.P2x is a variant of Sc.P2 known to not produce benzylpenicillin due an inactive *pcbAB* caused by mutation in the coding region of the gene's terminal module region. OD600 values from the bacterial cultures were

then measured again after overnight growth at 37 °C. The percentage growth inhibition caused by spent culture media for each strain was calculated by dividing the fold-change in OD600 overnight caused by that spent culture media by the overnight OD600 fold-change caused by the control media (from Sc.P2x cells) and subtracting this from 100%.

**Code availability.** Custom code for assigning nanopore sequencing reads to promoters in the first promoter screen (Fig. 2) is provided as part of Supplementary Data 3.

**Data availability.** Plasmid construct sequences are provided in Supplementary Data 1, LCMS spectra are available in Supplementary Data 2 and nanopore sequencing files are provided as Supplementary Data 3.

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

## Acknowledgements

We are grateful to Dr Verena Siewers and Professor Jens Nielsen for their donation of the pESC-npgA-pcbAB plasmid, and thank Professor Shiranee Sriskandan and Kristin Krohn Huse for their help preparing media and handling *S. pyogenes* for the growth inhibition assay. This work was funded in the UK by BBSRC awards BB/K006290/1 and BB/K019791/1 and EPSRC award EP/L011573/1

## Author contributions

A.R.A., B.A.B. and T.E. designed the study. A.R.A. performed all major experiments and did the statistical analyses. D.J.H. performed LCMS experiments. B.A.B. and R.M.M. performed nanopore sequencing. W.M.S. and J.C.H.H. provided additional experimental assistance. A.R.A. did all other experiments and performed all statistical analyses. A.R.A. and T.E. wrote the manuscript and B.A.B. and W.M.S. helped revise the manuscript.

## Additional information

**Competing interests:** The authors declare no competing financial interests.

**Publisher's note**: 

