## [Peer review file · Nature Communications]

Reviewers' comments:

Reviewer #1 (Remarks to the Author):

Awan, Ellis, et al. describe the production of the famous antibiotic penicillin (benzylpenicillin) in *Saccharomyces cerevisiae*. This work contains two main points of interest: 1) the first production of a non-ribosomal peptide using *S. cerevisiae* as a production host and 2) the pathway employs peroxisome compartmentalization for the last two enzymatic steps. I will comment on both of these main points below. Overall, the manuscript is deserving of publication if improved in a few points.

The first point is of importance because *S. cerevisiae* has superior synthetic biology tools for the production of other non-ribosomal peptides. For this particular product, another fungi, *Hansenula polymorpha*, has already been used for heterologous production. However, *S. cerevisiae* provides the tools for improvement, such as different strength promoters. A nice demonstration of a workflow for constructing a combinatorial expression library to screen for optimally balanced flux was demonstrated with the pathway. Higher expression levels were determined to be optimal: highest promoter strengths for genes *pclA* and *penDE* appear to be required for highest production (a control with a medium strength promoter should be done to validate this conclusion) while medium to highest strengths were enriched in the screen for *pbC*. Although not too different than other works with a similar aim of balancing flux through expression modulation, this part was well done and demonstrates the power of using *S. cerevisiae* as a host for future non-ribosomal peptide biosynthesis.

The second point of the importance of peroxisome compartmentalization in pathway performance more problematic as it is not as well described. Fig. 1c shows the peroxisome targeting of the biosynthetic enzymes, however the fraction of enzyme targeted versus in the cytoplasm is not clear and there looks to be ample enzyme in the cytoplasm. What expression levels are these, are they at the high expression levels observed to be optimum in Fig. 2b? Does the amount of enzyme in the peroxisome correlate with the increases in titer observed in the expression library? How does ACV gain access to the compartmentalized enzymes in the peroxisome? How well does the pathway function when these enzymes are not targeted to the peroxisome (i.e., all cytoplasmically expressed)?

Finally, the demonstration of bioactivity is well done. Although titers are very low (~5ug/L) the experiment described in Fig. 3 is an elegant demonstration that these titers are functional for the desired antibiotic bioactivity. The bacteria in Fig. 3c should be made a different color than the yeast cells for clarity. It would, though, be helpful to know how does the 5ug/L titer compare to what has been achieved in *Hansenula polymorpha*?

Reviewer #2 (Remarks to the Author):

The authors report the heterologous production of penicillin using *S. cerevisiae*, using synthetic biology approaches to improve titers and confirm bioactivity of product formed through the new host. Overall, the article is well-presented and fits nicely into a communication format. It is easy to read and the data presented are clear.

However, there are several items that are stopping me from being more supportive of publication.

First, is there a strong enough motivation to produce penicillin from yeast? The authors point to the engineering opportunities associated with *S. cerevisiae*, but it's unknown if similar opportunities could exist with the native host system or alternative fungal systems (such as *Aspergillus* spp.). Certainly, there would not seem to be a shortage of production from the original host, considering the effort that has gone into optimizing production over the years.

Second, I would classify the synthetic biology approaches used here as relatively standard (though effective). Therefore, unless these tools were able to extend production to the point where this new system could supplant current production for penicillin, I don't think their application is strong enough to support publication in *Nat. Comm.*

Finally, there is a question of novelty. The authors state that this is the first report of full benzylpenicillin produced through *S. cerevisiae*. They also note, commendably, that previous efforts have produced ACV. Furthermore, others have also used *S. cerevisiae* for nonribosomal peptide production previously. The use of *S. cerevisiae* for the production of this class of compounds, even if they are different from or not the full final compound targeted in this case, lessens the potential impact of the current work, in my opinion.

This isn't to lessen the quality of the work presented, however. I just feel that it would be more suitable for a more specialized journal given the points I raise above.

Reviewer #3 (Remarks to the Author):

Here, the authors reconstructed a complete pathway to benzylpenicillin in *S. cerevisiae*, including targeting two of the genes to the peroxisome to ensure correct functioning. They then combinatorially optimised production of the ACV intermediate by titrating expression of the two genes leading to its production using a promoter part library. Using a high ACV production strain, they then performed another combinatorial optimisation of the downstream pathway to penicillin (3 genes). In this case, they selected a subset of high/medium/low/inducible promoters and made a random combinatorial library of gene constructs, then selected based on phenotype and back-sequenced to identify the promoters in the constructs. Through this process, they were able to improve penicillin production 50-fold. A supernatant-based antibiotic screen demonstrated the antibiotic efficacy of the product and confirmed similar bioactivity compared with a commercial product.

The authors go on to discuss this as an example of the reconstruction of a complex fungal non-ribosomal peptide synthetase pathway to an industrially useful secondary compound in a GRAS model industrial organism, *S. cerevisiae*. They also discuss other applications for the technology in examination of NPRS enzymes and potentially identifying new antibiotics.

This is a very interesting piece of work and represents an excellent application of high-throughput combinatorial screening for pathway optimisation without the need for complex and expensive robotics. The target is a classical bio-product with very wide industrial use. The platform represents a novel contribution and has very exciting future potential. I have several comments with respect to the text and the story, as outlined below.

Major Comments

1. The novelty is not made as clear as it could be in the abstract or Introduction. The authors simply talk about correct subcellular localisation and pathway optimisation... but this may be the case in a lot of examples. What's so special about penicillin and/or the penicillin pathway? – here it is the use of the combinatorial promoter testing pipeline for optimisation of production for a complex pathway with support enzymes.

2. Figures 1b and 1d show HPLC traces for the intermediate ACV and benzylpenicillin, respectively. Ideally, all strains and the control should be included in all figures, and the traces should be next to each other so that the reader can see one peak disappear and a new one appear in response to the engineering. This may be in the same figure if they are analysed using the same method, or in separate figure parts if not.

3. Line 107-110: The library had a theoretical diversity of 1000 members but only 160 were screened by LC-MS. This is quite understandable given limitations in this analytical approach. However, this leaves quite a large portion of the generated solution space untested. The authors later demonstrate an absolutely facile screening technique that is readily applicable to development of high-throughput screening for thousands of strains: antimicrobial activity of culture supernatants. One could easily imagine a microtitre-based growth approach, spinning the plates and taking a sample of supernatant to dose into a second microtitre plate inoculated with the target organism, and simple spectrophotometric

analysis of growth rates to determine efficacy. Such an approach could perhaps be used to examine the remaining solution space and identify even better producers. Better yet (but perhaps beyond the scope of the current study), apply the combinatorial technique to all five genes in parallel and then use the supernatant testing technique to identify a super-producer.

4. In Figure 1c, there should be controls for subcellular localisation (see detailed comments below)

5. In the Discussion, a comparison to production of penicillin using current methods is required. If favourable, this warrants a mention in the Abstract

Specific Comments

1. Lines 63-67: reasoning for peroxisome localisation, and subsequent use of the PTS1 tag: there is a gap in logic flow of the story here which needs to be filled with some background information. Have the *pclA* and *penDE* genes been expressed previously in yeast and found to be inactive? The *pclA* and *penDE* genes presumably have a native peroxisome targeting sequence which works in the source organism. Do these targeting sequences not function in *S. cerevisiae*, or were they cleaved off during the cloning process? Or can they not be identified using current algorithms? Do the final constructs have both the native *P. chrysogenum* targeting sequences and the PTS1 tag on them? Also, worth mentioning the source of the PTS1 tag (presumably *S. cerevisiae*) and explicitly saying that this sequence has previously been shown to direct proteins to the *S. cerevisiae* peroxisome in previous studies (with appropriate reference).

2. Line 29: fungi should be fungus

3. Line 38: missing word; should be '...plethora of advanced...'

4. Line 73: would probably be fair to say 'very low' or 'tiny' here – helps provide the rationale for the high-throughput optimisation in the next section anyway

5. Line 80: to make clear the rationalisation and help emphasise the novelty of the current work, I suggest starting this section with a phrase something like, 'Considering the very tiny amounts of penicillin produced by the first generation engineered strain, we decided to optimise the production pathway. To do this, we developed a novel approach based on....' Or something like that.

6. Line 83: a reference for the statement about minimising build-up of inhibitory intermediates is required

7. Line 84: 'aid' is used twice in this sentence

8. Lines 85-88: a little more detail in the text about how hundreds of expression cassettes were designed, constructed, and tested would be good (including a few technical details of the extraction and analysis approach)

9. Line 88: reference to Fig 2 seems misplaced here. This figure describes the approach for optimisation of the second part of the pathway, however the rest of the paragraph described how the first part of the pathway was optimised.

10. Line 90: start a new paragraph from 'For the optimisation of...'

11. Lines 96-99: If the authors would like to make a statement attributing higher yields to increased enzyme expression, this must be substantiated by checking enzyme levels (e.g. using targeted proteomics). In fact, increased enzyme expression is not necessarily why final yields increase – in some cases, it is required to balance expression in a pathway to drive the most efficient catalytic process, and this may not be a 'more is better' equation. Furthermore, the statement about AT-rich sequences and inference that transcription is improved could alternatively be tested (but then the statement must be that increased transcription was observed, and that that presumably results in increased enzyme levels – which is not completely unreasonable, although of course translational efficiency will vary between genes). Note that in this case it's more likely that there is some kind of balance achieved through the high throughput combinatorial engineering; the GAL promoters are extraordinarily strong and I'd be surprised if the central carbon metabolism promoters out-competed them at the transcriptional level, unless some other transcriptional augmentation was included in the new constructs – as the authors seem to suggest. In this case, it's a bit disingenuous to attribute the improved activity to specific promoters alone. This augmentation should be specifically mentioned in the text as a key part of the design approach. Also note that the copy number of 2u plasmid can be altered by the yeast cell so if the yeast is experiencing increased metabolic load/toxicity from the construct, the copy number may be kept low. I'm not sure if this happens with other plasmids.

12. Line 109: the constructs were screened in YPD medium – but two of the promoters were galactose-inducible. Was the library perhaps also screened in galactose medium? Also note that this is the first time the medium is mentioned. It would be worthwhile mentioning earlier if a medium other than defined medium was used for analysis, since that's the only other medium mentioned previously in the text.

13. Line 110: 'media' (plural) should be 'medium' (singular)

14. Line 112: were only 10 of the 160 colonies producing detectable amounts of penicillin? Or were only 10 of the producing strains selected? If the latter, how many of the 160 produced detectable amounts? Please clarify, and explain selection criteria for these ten (e.g. maybe the top ten quantitative producers were selected).

15. Lines 115-116: over-representation of medium constitutive promoters at pcbC gene: I

would characterise this further and more specifically by saying that the highest producers primarily had strong promoters at *pclA* and *penDE* and medium-strength promoters at *pcbC*.

16. Line 134: state the starting point and end point for the >50-fold increase (x pg/mL to 5 ng/mL)

17. Line 136: 'media' should be 'medium'.

18. Line 141: please note in text here how inactivation was performed

Figure 1 caption:

19. Line 422: insert 'first' after '...and valine are...'

20. Line 428: 'were' should be 'was'

21. Line 436: what is the phenylacetic acid for? While supplementation using AAA is reasonably obvious, why add phenylacetic acid? This is the first place it's mentioned

Figure 1a

22. it would be worth representing the sub-cellular localisation of the pathway steps by e.g. drawing a yeast cell and annotating with the reaction locations

Figure 1b

23. Annotate the gene construct diagram with the pGAL promoters please. Also, it is unclear what the parallel angled lines denote in the *pcbAB* gene. Also, annotate terminator region names please.

24. mg/mL is a more commonly used unit (for ACV)

25. The traces are good to see, but I would also like to know that the mass spectrum also matched with the standard.

26. There seems no reason why this data can't be presented quantitatively next to the traces – i.e. generate a standard curve and determine the concentration of the ACV in the extracellular medium (NB – four standards, as described in the M&M, is insufficient for a proper standard curve. I would normally target eight, with five as a bare minimum, and be sure that the range of actual sample points is within the standard curve and that this range is well covered by the curve. Regression coefficient may be presented in the M&M along with a comment on how the curve was produced).

Figure 1c

27. Common controls for subcellular localisations should be included. Specifically, tagged versions of proteins known to be located in the cytosol and peroxisomes for direct comparison to the new tagged proteins. Or some other kind of location confirmation. Particularly for the peroxisomes – otherwise we are just looking at pretty spots and presuming they are peroxisomes

Figure 1d

28. As for Figure 1b with respect to annotation of figures, units for penicillin, and quantitative data/standard curves

29. I can see that the traces are plotted on different y-axes, as the yeast product is something like an order of magnitude lower than the standard. But why not just use a standard in the same range for the graphical representation?

30. It would be good to combine this figure with figure 1b, so that all strains (incl. WT, which is missing from this figure) and standards are shown in one figure. At the very least, the WT should be included in this figure – it is currently missing

31. Only one analytical method is described in the M&M so I presume it was used for both ACV and benzylpenicillin. Where has the ACV peak in strain Sc.A1 gone? There should be a peak just before 5 min. Ideally we should see a peak shift between Sc.A1 (ACV-producing) and Sc.P1 (penicillin-producing) to help indicate conversion of ACV to penicillin in the Sc.P1 strain. Additional quantitative presentation of the data would also give some indication of how complete the conversion is (presumably it is complete, since there is no ACV peak in Sc.P1. Also presumably this is due to the gene dosage, since the last 3 genes required are plasmid-expressed whereas the first two genes are integrated onto the chromosome; though it isn't mentioned in the text which promoters were used for the last three genes – hence the request to have the annotations in Fig 1d).

32. Notwithstanding the above comment, I note the following: while of course it may not be linear and it's the area under the curve not the peak height and of course different standards will behave differently, I note that in Fig 1b 100 pg/uL ACV provides a peak height of 6000 counts and Sc A1 gives 1/6th of this peak height (1000 counts); and just 10 pg/L penicillin gives 30,000 counts while Sc.P1 gives 1/30th of this (1000 counts). Thus, by rule of thumb, something is missing – I suspect that there is not full conversion of ACV into penicillin. So we should probably see an ACV peak just before 5 min for Sc.P1 in figure 1d.

33. Having now read the next sentence in the text as well as the M&M, I think I understand that the ACV was extracted from the cell pellet for figure 1b, and the penicillin came from the supernatant for figure 1d. While this clarifies the above confusion somewhat, it still holds that all strains should be assayed for both the ACV and the penicillin and all strains included on both figures, so it's clear which strains have which peaks, and so that one can determine if

full conversion occurs in the Sc.P1 strain. Also, it should be made clear in the figure caption that one product is collected from the supernatant and one from the cell pellet. For a product isolated from the supernatant, a mg/L unit is appropriate; for a product isolated from the cell biomass, it is probably more appropriate to present as mg/g DCW or something similar. Moreover, it appears that both products are actually present more or less in both the biomass and the medium, so probably the best way to present the data completely would be a supernatant figure with both ACV and penicillin, and a biomass figure with both ACV and penicillin.

34. After reading and making all of the above comments, I'm still confused as to why the ACV peak is missing from Sc.A1 in figure 1d. It is apparently present in both medium and supernatant. Perhaps it does not show up in the penicillin analytical method. NB – the analytical method for the ACV is not described in the M&M

Figure 2a:

35. for clarity, annotate the pcbAB / npgA cassettes on the yeast figure with the promoters. Again, the significance of the angled parallel lines in the pcbAB construct is unclear, but they appear to be in a different location than the lines in figure 1b and 1d.

Figure 3

36. Figure part 3a is cute, but probably unnecessary

Materials and Methods

37. the analytical method for the ACV is not described in the M&M

38. Lines 212-220: please include a description of the storage of strains after construction (how many replications, taken at which stage, etc.) and the recovery of strains from glycerol stocks prior to production analysis, including preculture strategy. This should be a very reproducible process, and include period of time incubated on solid medium, method of inoculation of pre-culture, whether or not there was a secondary preculture, at what OD it was cultured, and under what conditions precultures were grown, at what OD the production cultures inoculated, etc.

39. Lines 228-245: standards should be prepared densitometrically for accuracy – was this done? At least 5 standards in an appropriate range (covering all the target analyte concentrations) should be used to prepare a quantitative standard curve, and ideally a minimum of eight.

40. Lines 249-265 LC-MS/MS: use of penicillin-G instead of benzylpenicillin – please use the same terminology throughout the manuscript.

41. Lines 267-273: as per above, more standard concentrations required for quantitative standard curves

Thank you for providing us with a chance to improve our manuscript in order to address valuable suggestions from peer-review. Below we point-by-point address the comments made by each reviewer.

Note that as the first author of this work has since moved on from science, it has taken us slightly longer than we would have anticipated to respond. To accelerate things, we enlisted a new researcher (William Shaw) who carried out some of the suggested experimental changes. In recognition of his contribution he has now been included as an author in the new version of the manuscript.

Reviewer #1 (Remarks to the Author):

Awan, Ellis, et al. describe the production of the famous antibiotic penicillin (benzylpenicillin) in *Saccharomyces cerevisiae*. This work contains two main points of interest: 1) the first production of a non-ribosomal peptide using *S. cerevisiae* as a production host and 2) the pathway employs peroxisome compartmentalization for the last two enzymatic steps. I will comment on both of these main points below. Overall, the manuscript is deserving of publication if improved in a few points.

The first point is of importance because *S. cerevisiae* has superior synthetic biology tools for the production of other non-ribosomal peptides. For this particular product, another fungi, *Hansenula polymorpha*, has already been used for heterologous production. However, *S. cerevisiae* provides the tools for improvement, such as different strength promoters. A nice demonstration of a workflow for constructing a combinatorial expression library to screen for optimally balanced flux was demonstrated with the pathway. Higher expression levels were determined to be optimal: highest promoter strengths for genes *pclA* and *penDE* appear to be required for highest production (a control with a medium strength promoter should be done to validate this conclusion) while medium to highest strengths were enriched in the screen for *pbcC*. Although not too different than other works with a similar aim of balancing flux through expression modulation, this part was well done and demonstrates the power of using *S. cerevisiae* as a host for future non-ribosomal peptide biosynthesis.

We thank the reviewer for their positivity and in seeing the importance of working with this pathway in *S. cerevisiae* given the demonstrable tools available for engineering and optimisation in this organism.

Regarding the comment that a control with a medium strength promoter should be done to validate the conclusions of high strength expression being optimal - this is indeed a good point. However, experiments we performed to address other issues raised by reviewers have led us to now modify our earlier conclusions about promoter strength. Specifically, we have now performed a second promoter screen to highlight the use of our *S. pyogenes* growth inhibition assay. This second bioactivity-based screen identified strains with even higher yields than our first assay, and while this second screen also utilised strong promoters for the peroxisomal genes (Table S10), the very strongest promoters (e.g. TDH3) from this set were not always favoured (Fig 3d, Table S12), perhaps because gross overexpression of the enzymes is not the optimal strategy. In light of this we have now softened the language used in order to indicate that the exact relationship between benzylpenicillin production and the relative promoter strengths of the pathway genes is likely to be more complex than initially suggested by our first screen. This is perhaps not so surprising given that optimising metabolic pathways typically requires more consideration than simply overexpressing every enzyme.

The second point of the importance of peroxisome compartmentalization in pathway performance more problematic as it is not as well described.

Fig. 1c shows the peroxisome targeting of the biosynthetic enzymes, however the fraction of enzyme targeted versus in the cytoplasm is not clear and there looks to be ample enzyme in the cytoplasm. What expression levels are these, are they at the high expression levels

observed to be optimum in Fig. 2b? Does the amount of enzyme in the peroxisome correlate with the increases in titer observed in the expression library?

In light of these comments and those made by the third reviewer, we have now repeated our experimental analysis on peroxisome compartmentalization and provide new microscopy images now in Fig S1. These more clearly show that the localization is peroxisomal and dependent on the synthetic tags we have added to the enzymes. Note that in these images the promoters used to drive expression are strong promoters and are among those that are enriched in the screen shown in Fig 2b which determined optimal producers. Details of the promoters used are in the Figure legend.

With regards to estimating the fraction of enzyme that is cytoplasmic vs. peroxisomal, we considered simply measuring green channel intensity in our images in the cytoplasm vs peroxisome as a measurement method. Indeed, when doing this using ImageJ, we found that typically the green intensity levels in the peroxisomes were between 5 and 20 times greater than those in the cytoplasm, which were only just above background levels. However, we have decided not to include this information in our manuscript, as we don't believe that this kind of analysis is robust against imaging artifacts. For example, even though the yeast cells were fixed to the microscope slide, we observed that the peroxisomes move within the cells, in all three dimensions and as such the different focal planes for different peroxisomes affects the fluorescence intensity seen in the images. Without undertaking a whole new method of measurement it is difficult for us to determine the precise efficiency of translocation of the enzymes to the peroxisome and how it relates to promoter strength and resultant pathway performance.

How does ACV gain access to the compartmentalized enzymes in the peroxisome? Apologies, there seems to be a confusion here. ACV is not supposed to enter the peroxisome, instead ACV is first converted to isopenicillin N (IPN) by the activity of *pcbC* in the cytoplasm. In the original producer organism *P. chrysogenum*, it is thought that IPN is translocated from the cytoplasm to the peroxisome by active transport, involving a transmembrane transport protein called *penM* (Fernandez-Aguado *et al.* 2014). In *S. cerevisiae* how IPN is translocated to the peroxisome is unknown, and there is no apparent ortholog for *penM*, but transmembrane translocator proteins also exist in this organism (Andre 1995) which may provide an explanation. However, as this is only speculation at this stage we have chosen not to add this to the manuscript text.

How well does the pathway function when these enzymes are not targeted to the peroxisome (i.e., all cytoplasmically expressed)?

ACV does not get converted to benzylpenicillin in a strain where *pclA* and *penDE* are not targeted to peroxisomes. This has now been demonstrated and described in the updated version of Fig 1.

Finally, the demonstration of bioactivity is well done. Although titers are very low (~5ug/L) the experiment described in Fig. 3 is an elegant demonstration that these titers are functional for the desired antibiotic bioactivity. The bacteria in Fig. 3c should be made a different color than the yeast cells for clarity. It would, though, be helpful to know how does the 5ug/L titer compare to what has been achieved in *Hansenula polymorpha*?

We thank the reviewer for this suggestion. A comparison of our yields to other producers has now been added to the end of the first paragraph of the discussion. We have also changed the bacteria colour in Fig 3a.

Reviewer #2 (Remarks to the Author):

The authors report the heterologous production of penicillin using *S. cerevisiae*, using synthetic biology approaches to improve titers and confirm bioactivity of product formed through the new host. Overall, the article is well-presented and fits nicely into a communication format. It is easy to read and the data presented are clear.

However, there are several items that are stopping me from being more supportive of publication.

First, is there a strong enough motivation to produce penicillin from yeast? The authors point to the engineering opportunities associated with *S. cerevisiae*, but it's unknown if similar opportunities could exist with the native host system or alternative fungal systems (such as *Aspergillus* spp.). Certainly, there would not seem to be a shortage of production from the original host, considering the effort that has gone into optimizing production over the years.

We thank the reviewer for their thoughts on our manuscript, and apologise that the motivation of for our work was not clearly put forward. Producing penicillin at commercial yields from Baker's yeast has never been our goal, and this misunderstanding seems to be at the heart of comments here. As the reviewer correctly points out, existing production strategies in engineered variants of the natural producer fungus *P. chrysogenum* and via semi-synthetic chemistry approaches are already more-than sufficient for this antibiotic. Instead, the motivation for our work is two-fold; (1) to demonstrate that Baker's yeast can produce a bioactive Nrp by engineering both NRPS expression and the coordinated expression of tailoring enzymes, and (2) getting penicillin secretion to yields that can be quickly assayed by the ability for the cells to prevent bacterial growth. In achieving these goals we are laying the foundations for projects that we anticipate can mix and match heterologous enzymes and even NRPS modules/domains to create new bioactive molecules potentially with antibiotic properties. Work towards this is already underway in our lab and will form a separate future manuscript that builds upon the benchmark work described here.

Importantly, by doing this in *S. cerevisiae* we are significantly accelerating work towards biosynthesis of alternative Nrp molecules by benefiting from the many tools available for rapid and precise genetic engineering and for metabolic modelling in this model organism - tools that are lacking in *P. chrysogenum* and *Aspergillus* systems. As an added bonus, *S. cerevisiae* is extensively used in industry and so strains we generate will likely be more amenable to scale-up in the future and rational improvements by others.

In order to better clarify our motivation for this work, we have now added two sentences to the the second paragraph of the introduction that outline what we set out to demonstrate in this work.

Second, I would classify the synthetic biology approaches used here as relatively standard (though effective). Therefore, unless these tools were able to extend production to the point where this new system could supplant current production for penicillin, I don't think their application is strong enough to support publication in Nat. Comm. As mentioned above, supplanting current commercial production levels of penicillin is not the intended goal of this work and would be wholly unrealistic considering the decades of industrial process optimisation that has been achieved for this by the likes of DSM and GSK. Secondly, we disagree that the synthetic biology tools used in our work are standard. Indeed, the MoClo yeast combinatorial cloning system we utilised was only published last year and we are one of the first to use it for metabolic pathway optimisation. On top of this we are the first group to use Nanopore sequencing during combinatorial assembly to verify library diversity throughout construction, and this in itself is a novel advance. Tellingly, we have been approached by several DNA Foundry facilities in the past six months who are interested in implementing our Nanopore-based verification pipeline in their workflows.

In light of this comment we have now added two sentences to the manuscript discussion in order to clarify that the goal of this research is not to supplant current production, but to lay the foundations for modular engineering for the biosynthesis of new bioactive Nrp therapeutics.

Finally, there is a question of novelty. The authors state that this is the first report of full benzylpenicillin produced through *S. cerevisiae*. They also note, commendably, that previous efforts have produced ACV. Furthermore, others have also used *S. cerevisiae* for

nonribosomal peptide production previously. The use of *S. cerevisiae* for the production of this class of compounds, even if they are different from or not the full final compound targeted in this case, lessens the potential impact of the current work, in my opinion.

This isn't to lessen the quality of the work presented, however. I just feel that it would be more suitable for a more specialized journal given the points I raise above.

Biosynthesis of a nonribosomal peptide in Baker's yeast has only been demonstrated a handful of times, and as far as we are aware in all cases the work has simply been to produce an Nrp backbone molecule to be extracted/purified and used as substrate for *in vitro* chemical reactions and/or analysis. In no publications to date has the biosynthesis resulted in a bioactive molecule demonstrated to have antibiotic properties. Given the international urgency for solutions to antimicrobial resistance, we think that genetic engineering of complete biosynthesis (and secretion) of a bioactive clinically-used antibiotic by a widely-used organism is a novel achievement of broad interest beyond a specialist journal.

Reviewer #3 (Remarks to the Author):

Here, the authors reconstructed a complete pathway to benzylpenicillin in *S. cerevisiae*, including targeting two of the genes to the peroxisome to ensure correct functioning. They then combinatorially optimised production of the ACV intermediate by titrating expression of the two genes leading to its production using a promoter part library. Using a high ACV production strain, they then performed another combinatorial optimisation of the downstream pathway to penicillin (3 genes). In this case, they selected a subset of high/medium/low/inducible promoters and made a random combinatorial library of gene constructs, then selected based on phenotype and back-sequenced to identify the promoters in the constructs. Through this process, they were able to improve penicillin production 50-fold. A supernatant-based antibiotic screen demonstrated the antibiotic efficacy of the product and confirmed similar bioactivity compared with a commercial product.

The authors go on to discuss this as an example of the reconstruction of a complex fungal non-ribosomal peptide synthetase pathway to an industrially useful secondary compound in a GRAS model industrial organism, *S. cerevisiae*. They also discuss other applications for the technology in examination of NPRS enzymes and potentially identifying new antibiotics.

This is a very interesting piece of work and represents an excellent application of high-throughput combinatorial screening for pathway optimisation without the need for complex and expensive robotics. The target is a classical bio-product with very wide industrial use. The platform represents a novel contribution and has very exciting future potential. I have several comments with respect to the text and the story, as outlined below.

Major Comments

1. The novelty is not made as clear as it could be in the abstract or Introduction. The authors simply talk about correct subcellular localisation and pathway optimisation... but this may be the case in a lot of examples. What's so special about penicillin and/or the penicillin pathway? – here it is the use of the combinatorial promoter testing pipeline for optimisation of production for a complex pathway with support enzymes.

We thank the reviewer for their positive remarks about our research, our use of combinatorial high-throughput screening and the platform we have generated. As reviewer 2 also noted, the motivation for our research and its novelty was not clearly put forward in our first version of the manuscript. To address this we have now added two extra sentences to the introduction to outline what we set to demonstrate in this work; demonstrating that yeast can be engineered to produce a bioactive fungal Nrp molecule by coordinating expression of a NRPS and a pathway of tailoring enzymes.

2. Figures 1b and 1d show HPLC traces for the intermediate ACV and benzylpenicillin, respectively. Ideally, all strains and the control should be included in all figures, and the traces should be next to each other so that the reader can see one peak disappear and a new one

appear in response to the engineering. This may be in the same figure if they are analysed using the same method, or in separate figure parts if not.

We thank the reviewer for highlighting this suggested improvement to our figures. The updated manuscript now addresses this with separate figure parts shown in Figure 1 (c). These improvements to the figure also address the concerns raised in specific comments 29-34.

3. Line 107-110: The library had a theoretical diversity of 1000 members but only 160 were screened by LC-MS. This is quite understandable given limitations in this analytical approach. However, this leaves quite a large portion of the generated solution space untested. The authors later demonstrate an absolutely facile screening technique that is readily applicable to development of high-throughput screening for thousands of strains: antimicrobial activity of culture supernatants. One could easily imagine a microtitre-based growth approach, spinning the plates and taking a sample of supernatant to dose into a second microtitre plate inoculated with the target organism, and simple spectrophotometric analysis of growth rates to determine efficacy. Such an approach could perhaps be used to examine the remaining solution space and identify even better producers. Better yet (but perhaps beyond the scope of the current study), apply the combinatorial technique to all five genes in parallel and then use the supernatant testing technique to identify a super-producer.

Based on the suggestions of the reviewer we have now performed a second promoter screen demonstrating how the bioactivity assay can be exploited as a screening tool. This new round of screening explored a slightly modified promoter space consisting of six promoters: 4 strong and 2 medium strength (Table S9). This screen was carried out using the bioactivity assay, and did indeed identify better benzylpenicillin producers (see Fig 3). Extra text has now been added to the end of the results section of the manuscript to describe this.

4. In Figure 1c, there should be controls for subcellular localisation (see detailed comments below)

We have now added a new supplementary figure (Fig S1) with two sets of controls that conclusively demonstrate the peroxisomal localisation of *pclA* and *penDE*. One control is a fluorescently-tagged known peroxisomal protein, *CIT2*. Co-localisation of *CIT2* with *pclA* and *penDE* is demonstrated in Fig S1. The second control shows that the same fluorescently tagged proteins (*CIT2*, *pclA* and *penDE*) do not localise to peroxisomes in a strain in which the peroxisomal transport protein *pex5* is deleted. These new data were also generated in order to address specific comment 27 below.

5. In the Discussion, a comparison to production of penicillin using current methods is required. If favourable, this warrants a mention in the Abstract

We have now added a comparison of our yields to *P. chrysogenum* yields within the first paragraph of the discussion section. Note that this is also now followed by a sentence clarifying that achieving high-yield production is not a priority goal for this work (see response to Reviewer 2)

Specific Comments

1. Lines 63-67: reasoning for peroxisome localisation, and subsequent use of the PTS1 tag: there is a gap in logic flow of the story here which needs to be filled with some background information. Have the *pclA* and *penDE* genes been expressed previously in yeast and found to be inactive?

We have now shown that *pclA* and *penDE* with the *P. chrysogenum* PTS1 tag and without the *S. cerevisiae* PTS1 tag do not get targeted to *S. cerevisiae* peroxisomes (Fig S1) and that a strain with all five penicillin pathway genes but with *P. chrysogenum* PTS1 tags instead of *S. cerevisiae* PTS1 tags are able to produce ACV but not benzylpenicillin (Fig 1c).

The *pclA* and *penDE* genes presumably have a native peroxisome targeting sequence which works in the source organism. Do these targeting sequences not function in *S. cerevisiae*, or were they cleaved off during the cloning process? Or can they not be identified using current algorithms?

The new data shown in Fig S1 now demonstrate that the native peroxisomal targeting sequences from *P. chrysogenum* do not function well in *S. cerevisiae* as *pclA* or *penDE* are not efficiently located in the peroxisomes.

Do the final constructs have both the native *P. chrysogenum* targeting sequences and the PTS1 tag on them?

As described in the methods section “Construction of strains”, the final constructs for benzylpenicillin-producing strains only have *S. cerevisiae* PTS1 tripeptide tags (SKL) on them. In both cases, this replaces the native *P. chrysogenum* versions of this tag (SKI for *pclA* and ARL for *penDE*).

Also, worth mentioning the source of the PTS1 tag (presumably *S. cerevisiae*) and explicitly saying that this sequence has previously been shown to direct proteins to the *S. cerevisiae* peroxisome in previous studies (with appropriate reference).

We have now added the source of the PTS1 tag to the manuscript, in the section of the results with the heading “Establishing biosynthesis and secretion of benzylpenicillin in *S. cerevisiae*”, in the sentence “We therefore took the step of tagging both *pclA* and *penDE* with the previously characterised *S. cerevisiae* peroxisome targeting sequence (PTS1) tag”, which now has a reference to the study that demonstrates the targeting properties of this tag (Gould, S.J., Keller, G.A., Hosken, N., Wilkinson, J. & Subramani, S. *J Cell Biol* 108, 1657-1664 (1989).)

2. Line 29: fungi should be fungus

Corrected

3. Line 38: missing word; should be ‘...plethora of advanced...’

Corrected

4. Line 73: would probably be fair to say ‘very low’ or ‘tiny’ here – helps provide the rationale for the high-throughput optimisation in the next section anyway

Corrected

5. Line 80: to make clear the rationalisation and help emphasise the novelty of the current work, I suggestion starting this section with a phrase something like, ‘Considering the very tiny amounts of penicillin produced by the first generation engineered strain, we decided to optimise the production pathway. To do this, we developed a novel approach based on....’ Or something like that.

We thank the reviewer for the suggestion. This sentence now reads... “Considering the small amounts of benzylpenicillin produced by the first engineered strain Sc.P1, we next sought to optimise production of benzylpenicillin in *S. cerevisiae* in order to achieve bioactive concentrations.”

6. Line 83: a reference for the statement about minimising build-up of inhibitory intermediates is required

A reference has now been added: “F.Y. Lim, N.P. Keller *Nat. Prod. Rep.*, 31 (2014), pp. 1277–1286”

7. Line 84: ‘aid’ is used twice in this sentence

Now corrected

8. Lines 85-88: a little more detail in the text about how hundreds of expression cassettes were designed, constructed, and tested would be good (including a few technical details of the extraction and analysis approach)

We have now added a line to indicate that we used Golden Gate combinatorial DNA assembly for the construction of our library, and we have provided a reference to indicate how this approach works: “Engler, C., Kandzia, R., and Marillonnet, S. (2008) *PLoS ONE* 3, e3647.”

9. Line 88: reference to Fig 2 seems misplaced here. This figure describes the approach for

optimisation of the second part of the pathway, however the rest of the paragraph described how the first part of the pathway was optimised.

We thank the reviewer for noticing this omission. We had forgotten to include a reference to figure S1 (now Figure S2) which deals with the optimisation of the first part of the pathway. This is now corrected.

10. Line 90: start a new paragraph from 'For the optimisation of....'

This has been corrected

11. Lines 96-99: If the authors would like to make a statement attributing higher yields to increased enzyme expression, this must be substantiated by checking enzyme levels (e.g. using targeted proteomics). In fact, increased enzyme expression is not necessarily why final yields increase – in some cases, it is required to balance expression in a pathway to drive the most efficient catalytic process, and this may not be a 'more is better' equation.

We thank the reviewer for drawing our attention to these points which are all valid and capture the complexity of trying to attribute increased pathway productivity to the many possible changes in genetic design. In the updated manuscript we have now softened the language used in order to now convey that the exact relationship between benzylpenicillin production and the relative promoter strengths of the pathway genes is likely to be more complex than initially suggested by our first screen. As the reviewer points out, this is perhaps not so surprising given that optimising metabolic pathways typically requires more consideration than simply overexpressing every enzyme. Indeed, the new experiments we performed to address other issues raised by reviewers have now led us to modify our earlier conclusions about promoter strength. The bioactivity-based screen we have now added identified strains with even higher yields than our first assay, and while this second screen also utilised strong promoters for the peroxisomal genes (Fig 3d, Table S9), the very strongest promoters (e.g. TDH3) from this set were not always favoured, perhaps because gross overexpression of the enzymes is not the optimal strategy.

Furthermore, the statement about AT-rich sequences and inference that transcription is improved could alternatively be tested (but then the statement must be that increased transcription was observed, and that that presumably results in increased enzyme levels – which is not completely unreasonable, although of course translational efficiency will vary between genes). Note that in this case it's more likely that there is some kind of balance achieved through the high throughput combinatorial engineering; the GAL promoters are extraordinarily strong and I'd be surprised if the central carbon metabolism promoters out-competed them at the transcriptional level, unless some other transcriptional augmentation was included in the new constructs – as the authors seem to suggest. In this case, it's a bit disingenuous to attribute the improved activity to specific promoters alone. This augmentation should be specifically mentioned in the text as a key part of the design approach. Also note that the copy number of 2u plasmid can be altered by the yeast cell so if the yeast is experiencing increased metabolic load/toxicity from the construct, the copy number may be kept low. I'm not sure if this happens with other plasmids.

In the updated manuscript we have now removed the statement about AT-rich sequences that is highlighted here. However to clarify for the benefit of the reviewer, our suggestion regarding the AT-rich sequences was that the presence of these in the 5'UTR close to the start codon could be significantly enhancing translation from the ACVS and *npgA* mRNAs. At this region in the 5'UTR (sometimes called the Kozak sequence), AT-rich RNA sequences boost translation initiation in yeast compared to GC-rich regions (see Dvir, S. et al. 2013 PNAS USA 110). The previously described pESC-*npgA*-*pcbAB* construct had significant GC-content in this location which may have lowered translation efficiency from its mRNAs and therefore led to worse enzyme expression from this construct despite it having stronger promoters (GAL1) and higher copy-number (2u plasmid) than our new constructs. Again, another potential example of the interplay between different genetic design elements and resulting enzyme levels.

12. Line 109: the constructs were screened in YPD medium – but two of the promoters were galactose-inducible. Was the library perhaps also screened in galactose medium? Also note that this is the first time the medium is mentioned. It would be worthwhile mentioning earlier if a medium other than defined medium was used for analysis, since that's the only other medium mentioned previously in the text.

We thank the reviewer for pointing this out: this was an error. Strains were never screened in YPD but rather in Synthetic defined medium with glucose as the carbon source, minus the appropriate amino acids for selection purposes. The text of the manuscript has now been amended to reflect this, at the end of paragraph 3 of the section with the title: "Pathway optimisation with combinatorial library analysis by nanopore sequencing"

We initially included galactose-inducible promoters in the combinatorial assembly with a view to using them in Gal media. However, as the best yields of our ACV Nrp molecule from the first part of the pathway were always in glucose we reasoned it was pointless to test in Synthetic Defined medium plus Galactose. Instead, we kept the galactose-inducible promoters in the assembly to act as negative controls / weak promoters not expected to be present in assemblies giving the highest-yields of penicillin. Note that the uninduced GAL1 promoter is still expected to produce some gene expression, albeit at a very low level.

13. Line 110: 'media' (plural) should be 'medium' (singular)

Corrected

14. Line 112: were only 10 of the 160 colonies producing detectable amounts of penicillin? Or were only 10 of the producing strains selected? If the latter, how many of the 160 produced detectable amounts? Please clarify, and explain selection criteria for these ten (e.g. maybe the top ten quantitative producers were selected).

The total number of selected colonies producing detectable amounts of penicillin was 27 out of 160, and the top 10 of these were chosen for sequence analysis. These numbers and the criteria for choosing the top ten are now specified in the text of the revised manuscript.

15. Lines 115-116: over-representation of medium constitutive promoters at pcbC gene: I would characterise this further and more specifically by saying that the highest producers primarily had strong promoters at pclA and penDE and medium-strength promoters at pcbC. We thank the reviewer for this comment. In the interest of rigour we only specified those over-representations that were statistically significant (statistics calculated in Table S3).

16. Line 134: state the starting point and end point for the >30-fold increase (x pg/mL to 3 ng/mL)

The text has now been amended to include the starting point and the ending point.

17. Line 136: 'media' should be 'medium'.

Corrected

18. Line 141: please note in text here how inactivation was performed

Corrected

Figure 1 caption:

19. Line 422: insert 'first' after '...and valine are...'

Corrected

20. Line 428: 'were' should be 'was'

Corrected

21. Line 436: what is the phenylacetic acid for? While supplementation using AAA is reasonably obvious, why add phenylacetic acid? This is the first place it's mentioned

We have now added the following text after phenylacetic acid... "(required for the last step of benzylpenicillin biosynthesis)"

Figure 1a

22. it would be worth representing the sub-cellular localisation of the pathway steps by e.g. drawing a yeast cell and annotating with the reaction locations

We appreciate this suggestion, however we do not think that this is necessary given the textual description of the pathway, and it also risks complicating the clarity of an already busy figure for the reader.

Figure 1b

23. Annotate the gene construct diagram with the pGAL promoters please. Also, it is unclear what the parallel angled lines denote in the pcbAB gene. Also, annotate terminator region names please.

We thank the reviewer for these suggestions. We have now annotated the promoters in Fig 1b.

To explain the parallel angled lines, we have added the following text in the legend for Fig 1b: "Parallel diagonal lines in the cartoon for the pcbAB gene represent that this very large gene is not drawn to scale with npgA."

24. mg/mL is a more commonly used unit (for ACV)

We have now updated the figure to make the units for the standards "/mL" rather than "/µL"

25. The traces are good to see, but I would also like to know that the mass spectrum also matched with the standard.

We thank the reviewer for this suggestion. We have now included the mass spectra for the traces in Figure 1c as a zip file called "Supplemental spectra". This file is referred to in methods section and the legend for Fig 1c: "Mass spectra are included as a supplementary zip file".

26. There seems no reason why this data can't be presented quantitatively next to the traces – i.e. generate a standard curve and determine the concentration of the ACV in the extracellular medium (NB – four standards, as described in the M&M, is insufficient for a proper standard curve. I would normally target eight, with five as a bare minimum, and be sure that the range of actual sample points is within the standard curve and that this range is well covered by the curve. Regression coefficient may be presented in the M&M along with a comment on how the curve was produced).

We thank the reviewer for this point, but as this figure is intended to focus on demonstrating detectable production rather than the actual yields, we don't think it is necessary to convert the data into quantitative format at this stage, although it is a nice suggestion.

In terms of the number of standards - yes, it is regretful that we did not perform each of our experiments with a larger number of standards from the outset. We were confident that our four standards would suffice. For new experiments carried out in response to reviewer concerns we have used six-point standards. However, since at this stage we cannot repeat all of our previous experiments again with new standards, we instead chose to investigate the reviewer's comment by comparing our four-point standard curve (shown in red below) with a newly-done eight point standard curve (shown in blue below) for ACV, from a range spanning <1 ng/mL to 10,000 ng/mL.

As the graph clearly shows, there is a very strong match between the two curves. We note that the R-squared value for both standard curves is > 0.999 . When each curve is used to quantify a point on the other curve, the average error is under 5%. This gives us confidence that our four-point curve approach is valid and that repeating all our LCMS data with new standards should not be required.

A further important point for consideration is that our biological assay for detecting benzylpenicillin in culture media from our strain Sc.P2 quantifies the yield at ~ 3 ng/mL, which is almost exactly the amount we calculated using a four point standard curve from LCMS (3.05 ng/mL). This further supports our confidence with the four-point curves we have used, although we of course acknowledge that eight points would've been better.

Figure 1c

27. Common controls for subcellular localisations should be included. Specifically, tagged versions of proteins known to be located in the cytosol and peroxisomes for direct comparison to the new tagged proteins. Or some other kind of location confirmation. Particularly for the peroxisomes – otherwise we are just looking at pretty spots and presuming they are peroxisomes

We thank the reviewer for this suggestion. We now have a supplemental figure (Fig S1) that demonstrates peroxisomal localisation for *pclA* and *penDE* with two different sets of controls. Please also see the response to major comment 4 above.

Figure 1d

28. As for Figure 1b with respect to annotation of figures, units for penicillin, and quantitative data/standard curves

This figure panel is now incorporated into Figure 1c as per the suggestion of the reviewer.

29. I can see that the traces are plotted on different y-axes, as the yeast product is something like an order of magnitude lower than the standard. But why not just use a standard in the same range for the graphical representation?

We have now done this in the updated Fig 1c version.

30. It would be good to combine this figure with figure 1b, so that all strains (incl. WT, which is missing from this figure) and standards are shown in one figure. At the very least, the WT should be included in this figure – it is currently missing

We have now done this in the updated Fig 1c version.

31. Only one analytical method is described in the M&M so I presume it was used for both ACV and benzylpenicillin. Where has the ACV peak in strain Sc.A1 gone? There should be a peak just before 5 min. Ideally we should see a peak shift between Sc.A1 (ACV-producing) and Sc.P1 (penicillin-producing) to help indicate conversion of ACV to penicillin in the Sc.P1 strain. Additional quantitative presentation of the data would also give some indication of how complete the conversion is (presumably it is complete, since there is no ACV peak in Sc.P1. Also presumably this is due to the gene dosage, since the last 3 genes required are plasmid-expressed whereas the first two genes are integrated onto the chromosome; though it isn't mentioned in the text which promoters were used for the last three genes – hence the request to have the annotations in Fig 1d).

32. Notwithstanding the above comment, I note the following: while of course it may not be linear and it's the area under the curve not the peak height and of course different standards will behave differently, I note that in Fig 1b 100 pg/uL ACV provides a peak height of 6000 counts and Sc A1 gives 1/6th of this peak height (1000 counts); and just 10 pg/L penicillin gives 30,000 counts while Sc.P1 gives 1/30th of this (1000 counts). Thus, by rule of thumb, something is missing – I suspect that there is not full conversion of ACV into penicillin. So we should probably see an ACV peak just before 5 min for Sc.P1 in figure 1d.

33. Having now read the next sentence in the text as well as the M&M, I think I understand that the ACV was extracted from the cell pellet for figure 1b, and the penicillin came from the supernatant for figure 1d. While this clarifies the above confusion somewhat, it still holds that all strains should be assayed for both the ACV and the penicillin and all strains included on both figures, so it's clear which strains have which peaks, and so that one can determine if full conversion occurs in the Sc.P1 strain. Also, it should be made clear in the figure caption that one product is collected from the supernatant and one from the cell pellet. For a product isolated from the supernatant, a mg/L unit is appropriate; for a product isolated from the cell biomass, it is probably more appropriate to present as mg/g DCW or something similar. Moreover, it appears that both products are actually present more or less in both the biomass and the medium, so probably the best way to present the data completely would be a supernatant figure with both ACV and penicillin, and a biomass figure with both ACV and penicillin.

34. After reading and making all of the above comments, I'm still confused as to why the ACV peak is missing from Sc.A1 in figure 1d. It is apparently present in both medium and supernatant. Perhaps it does not show up in the penicillin analytical method. NB – the analytical method for the ACV is not described in the M&M

We thank the reviewer for the above comments (31-34) interpreting the Figure data and detailing suggested improvements for Figure 1. To address these issues we have now presented the peaks for ACV and for benzylpenicillin side by side in Figure 1c as suggested which hopefully gives a clearer picture. Note that these are on two different graphs due to the vastly different LCMS counts for both molecules which requires different y-axis scales. In these graphs we show all standards and samples together in a much simpler format compared to the previous version. This graph shows a repeat of the experiment used to generate the data for the last version of Figure 1C, but now with more samples being tested for ACV and benzylpenicillin titres.

Note that our method is only measuring ACV and benzylpenicillin, not isopenicillin N (an intermediate between ACV and benzylpenicillin). The absence of this in our detection could account for some of the differences between ACV and benzylpenicillin peaks and help explain what may be missing in comment 32.

Figure 2a:

35. for clarity, annotate the pcbAB / npgA cassettes on the yeast figure with the promoters. Again, the significance of the angled parallel lines in the pcbAB construct is unclear, but they appear to be in a different location than the lines in figure 1b and 1d.

We thank the reviewer for these comments. The promoter labelling has now been done. The parallel lines serve the same purpose as in figure 1 (rendering their location in the gene cartoon irrelevant, please see the answer to point 23 above) and since they were explained in the legend for that figure, we have omitted repeating the explanation in the legend for figure 2.

Figure 3

36. Figure part 3a is cute, but probably unnecessary

We thank the reviewer for this observation, but we would like to keep part 3a as we feel that it aids in the understanding of how the screen was set up.

Materials and Methods

37. the analytical method for the ACV is not described in the M&M

We thank the reviewer for the comment, it was not made clear that the method described in the "LCMS" section of the materials and methods applies to both benzylpenicillin and ACV. Accordingly, we have changed the sentence that used to read "An LC/MS/MS method was developed for the measurement of benzylpenicillin and related materials, using an Agilent 1290 LC and 6550 quadrupole time-of-flight (Q-ToF) mass spectrometer with electrospray ionization (Santa Clara, CA)." to "An LC/MS/MS method was developed for the measurement of ACV and benzylpenicillin, using an Agilent 1290 LC and 6550 quadrupole time-of-flight (Q-ToF) mass spectrometer with electrospray ionization (Santa Clara, CA)."

Further, we have added a line referring readers to table S6, where further details for ACV and benzylpenicillin LCMS are given.

38. Lines 212-220: please include a description of the storage of strains after construction (how many replications, taken at which stage, etc.) and the recovery of strains from glycerol stocks prior to production analysis, including preculture strategy. This should be a very reproducible process, and include period of time incubated on solid medium, method of inoculation of pre-culture, whether or not there was a secondary preculture, at what OD it was cultured, and under what conditions precultures were grown, at what OD the production cultures inoculated, etc.

We thank the reviewer for pointing this out. This has now been done in the section with the heading: "Growth of Strains for ACV and penicillin production".

39. Lines 228-245: standards should be prepared densitometrically for accuracy – was this done? At least 5 standards in an appropriate range (covering all the target analyte concentrations) should be used to prepare a quantitative standard curve, and ideally a minimum of eight.

Please see the response to specific comment #26 above.

40. Lines 249-265 LC-MS/MS: use of penicillin-G instead of benzylpenicillin – please use the same terminology throughout the manuscript.

This has now been corrected.

41. Lines 267-273: as per above, more standard concentrations required for quantitative standard curves

Please see the response to specific comment #26 above.

REVIEWERS' COMMENTS:

Reviewer #1 (Remarks to the Author):

I believe the manuscript has been improved, particularly with the changes to more clearly frame the purpose of the work demonstrating *S. cerevisiae* as a viable production host for production of NRP antibiotics. The authors addressed the questions I had. In particular, it is satisfying to know that ACV is not converted to benzylpenicillin in a strain where *pclA* and *penDE* are localized in the cytoplasm. This eases my concern about the necessity of peroxisome targeting. The manuscript is now clearer and I would be favorable for publication.

Some minor writing edits:

Line 177. Change media to medium

Line 177-179. Fix sentence. E.g. change to "...and selected 12 strains that exhibited a range of *S. pyogenes* inhibition..."

Line 189-191. Fix sentence. E.g. change to "Encouragingly, the results from our screen illustrate...."

Reviewer #2 (Remarks to the Author):

I appreciate the responses of the authors and, in particular, the effort put forth in addressing experimental questions by other reviewers. However, I am still struggling with the degree of novelty provided in this work.

The authors respond that the eventual goal of this work is not to supplant current penicillin production from *P. chrysogenum* but instead to lay the foundation for future NRP production from yeast. However, my argument is that because others have produced penicillin from yeast systems and nonribosomal peptides from *S. cerevisiae*, the current work does not meet a degree of novelty necessary for an article in *Nat. Comm.*

In a similar line of thinking, the optimization strategies resemble several previous efforts of promoter engineering and similar techniques, even if various elements used are relatively new.

Again, I want to emphasize that this is good work, but it is unclear whether a necessary novelty threshold has been crossed for publication in a journal like *Nat. Comm.*

Reviewer #3 (Remarks to the Author):

The authors have done an excellent job of answering the concerns of the reviewers. I have no further comments

Best regards,
Claudia Vickers - a.k.a. Reviewer 3

Below we point-by-point address the comments made by each reviewer.

Reviewer #1 (Remarks to the Author):

I believe the manuscript has been improved, particularly with the changes to more clearly frame the purpose of the work demonstrating *S. cerevisiae* as a viable production host for production of NRP antibiotics. The authors addressed the questions I had. In particular, it is satisfying to know that ACV is not converted to benzylpenicillin in a strain where *pclA* and *penDE* are localized in the cytoplasm. This eases my concern about the necessity of peroxisome targeting. The manuscript is now clearer and I would be favorable for publication.

We thank the reviewer for their positive comments and also for identifying the below edits.

Some minor writing edits:

Line 177. Change media to medium
This has now been corrected

Line 177-179. Fix sentence. E.g. change to "...and selected 12 strains that exhibited a range of *S. pyogenes* inhibition..."
This has now been corrected

Line 189-191. Fix sentence. E.g. change to "Encouragingly, the results from our screen illustrate..."
This has now been corrected

Reviewer #2 (Remarks to the Author):

I appreciate the responses of the authors and, in particular, the effort put forth in addressing experimental questions by other reviewers. However, I am still struggling with the degree of novelty provided in this work.

The authors respond that the eventual goal of this work is not to supplant current penicillin production from *P. chrysogenum* but instead to lay the foundation for future NRP production from yeast. However, my argument is that because others have produced penicillin from yeast systems and nonribosomal peptides from *S. cerevisiae*, the current work does not meet a degree of novelty necessary for an article in Nat. Comm.

In a similar line of thinking, the optimization strategies resemble several previous efforts of promoter engineering and similar techniques, even if various elements used are relatively new.

Again, I want to emphasize that this is good work, but it is unclear whether a necessary novelty threshold has been crossed for publication in a journal like Nat. Comm.

We are glad to hear that the reviewer now sees no technical issues or faults with our work or manuscript. However, we disagree that our work is not sufficiently novel to justify publication in this journal, and we are happy that both of the other reviewers share our point of view. Considering that *S. cerevisiae* is the most extensively-used and engineered organism for biosynthesis projects and yet has never before been engineered to produce a bioactive NRPS drug, in our view a sign that this is a major achievement. Given the current interests in both metabolic engineering and antibiotics research, we feel that our work is worthy of publication to a broad-audience.

Reviewer #3 (Remarks to the Author):

The authors have done an excellent job of answering the concerns of the reviewers. I have no further comments

Best regards,
Claudia Vickers - a.k.a. Reviewer 3

We thank the reviewer for their positive comments and for helping us improve the manuscript through peer review.